# In Vivo Inhibitory Assessment of Potential Antifungal Agents on *Nosema ceranae* Proliferation in Honey Bees

**DOI:** 10.3390/pathogens11111375

**Published:** 2022-11-18

**Authors:** Rassol Bahreini, Medhat Nasr, Cassandra Docherty, Olivia de Herdt, David Feindel, Samantha Muirhead

**Affiliations:** 1Plant and Bee Health Surveillance Section, Alberta Agriculture and Irrigation, Edmonton, AB T5Y 6H3, Canada; 2Department of Biological Sciences, University of Alberta, Edmonton, AB T6G 2E9, Canada; 3Saskatchewan Beekeepers Development Commission, Prince Albert, SK S6V 6Z2, Canada

**Keywords:** *Apis mellifera*, *Nosema ceranae*, fenbendazole, ornidazole, Fumagilin-B^®^

## Abstract

*Nosema ceranae* Fries, 1996, causes contagious fungal nosemosis disease in managed honey bees, *Apis mellifera* L. It is associated around the world with winter losses and colony collapse disorder. We used a laboratory in vivo screening assay to test curcumin, fenbendazole, nitrofurazone and ornidazole against *N. ceranae* in honey bees to identify novel compounds with anti-nosemosis activity compared to the commercially available medication Fumagilin-B^®^. Over a 20-day period, *Nosema*-inoculated bees in Plexiglas cages were orally treated with subsequent dilutions of candidate compounds, or Fumagilin-B^®^ at the recommended dose, with three replicates per treatment. Outcomes indicated that fenbendazole suppressed *Nosema* spore proliferation, resulting in lower spore abundance in live bees (0.36 ± 1.18 million spores per bee) and dead bees (0.03 ± 0.25 million spores per bee), in comparison to Fumagilin-B^®^-treated live bees (3.21 ± 2.19 million spores per bee) and dead bees (3.5 ± 0.6 million spores per bee). Our findings suggest that Fumagilin-B^®^ at the recommended dose suppressed *Nosema*. However, it was also likely responsible for killing *Nosema*-infected bees (24% mortality). Bees treated with fenbendazole experienced a greater survival probability (71%), followed by ornidazole (69%), compared to *Nosema*-infected non-treated control bees (20%). This research revealed that among screened compounds, fenbendazole, along with ornidazole, has potential effective antifungal activities against *N. ceranae* in a controlled laboratory environment.

## 1. Introduction

A number of organisms affect the health of European honey bees *Apis mellifera* L. A group of spore-forming obligate fungal endo-parasites, *Nosema* spp., is a major honey bee disease and contributes to colony losses worldwide. Three known species of *Nosema* exist that parasitize *A. mellifera* including *Nosema ceranae* Fries, 1996, *Nosema apis* Zander, 1909, and most recently discovered, *Nosema neumanni* n. sp., from Uganda [1]. Tokarev et al. [2] reclassified *Nosema* genera and recommended *Vairimorpha ceranae* and *Vairimorpha apis* as new scientific names for *N. ceranae* and *N. apis*, respectively. However, henceforth, we will refer to *Varimorpha* as *Nosema* to remain consistent with other literature. It is believed that *Nosema apis* co-evolved with *A. mellifera* [3] and was first described in Europe in the early 1900s [4]. After the introduction of *A. mellifera* to North America, *N. apis* remained the dominant species until *N. ceranae* was detected in early 2007 [5,6]. Although *N. apis* and *N. ceranae* can coexist in some populations [7,8,9], a recent study found that *N. ceranae* has become the dominant species of *Nosema* in Canada [10].

*Nosema* affects all castes found in a honey bee colony. Infections are primarily transmitted throughout the bee population by trophallaxis and there is evidence that both *N. ceranae* and *N. apis* can be sexually transmitted to a healthy queen through mating [11]. *Nosema* is an obligate intracellular parasites that targets the epithelial cells of the adult and larvae midgut [12]. Damage to the epithelium tissue can trigger energetic stress in the host, cause loss of energy that can reduce the longevity of honey bees, affect early spring colony build-up, and increase winter mortality [9,13,14]. *Nosema* infections can affect honey bee behavior (i.e., grooming), physiology, and immune gene expression [15,16,17,18,19]. The changes in gene expression can lead to a reduction in vitellogenin and an increase in juvenile hormone titers in infected bees, resulting in premature foraging [17,18].

To prevent infection, beekeepers have relied on a single anti-microbial fungicide, fumagillin, for more than 70 years [20,21]. Fumagillin is one of a few active ingredients available for treatment of *Nosema* spp. in Canada. Fumagilin-B^®^ (Medivet Pharmaceuticals, Alberta, Canada: currently Vita Bee Health, Basingstoke, UK) and Fumidil-B^®^ (currently KBNP Inc., Chungnam, South Korea) are two commercially available products containing the dicyclohexylamine salt of fumagillin [22]. Fumagillin targets methionine aminopeptidase type 2 that is present in *Nosema* spp., bees, and humans [23,24]. Although studies have been conducted into the effectiveness of fumagillin on *N. apis* [25,26], it is still unclear what impact fumagillin has on *N. ceranae* [14,25]. Low efficacy of Fumagilin-B^®^ on *N. ceranae* has been reported in laboratory experiments when *Nosema*-infected bees were exposed to lower dilutions than recommended [26]. Evidence exists that fumagillin may be harmful to honey bees [22,26,27]. Resistance to fumagillin has not been reported in either *N. apis* or *N. ceranae,* but has been documented in other *Nosema* species that infect insects [28]. Fumagillin has also been used to treat amoebiasis in humans [29]. Some side- effects in mammals have been documented such as chromosomal aberrations, genotoxic potential, gastrointestinal cramping, diarrhea, and significant weight loss [30,31]. Because of fumagillin’s potential toxic effect on humans, and the risk of residues in honey and wax, its use has been banned in the European Union [32], Australia [33] and South America [34].

Over-reliance on fumagillin as a primary treatment for *Nosema* presents challenges around its effectiveness against *N. ceranae*, the potential development of resistance, and the possible effect on bee and human health. With few highly effective treatment options, new products are required to ensure that beekeepers have tools available to keep honey bee colonies healthy. Numerous investigations have evaluated plant-based extractions, prebiotics and probiotics, and synthetic compounds against *Nosema* spp., all with varying results e.g., [22,32,34,35]. Anti-nosemosis activity of plant extracts on *Nosema* spp. and *A. mellifera* have been observed using caffeine, kaempferol, resveratrol, thymol, and essential oils of *Aster scaber* Thunb, *Artemisia dubia* Wall, *Andrographis paniculate* (Burm.f.) Nees, and *Eleutherococcus senticosus* (Rupr. et Maxim.) Maxim. and Oliv [36,37,38,39,40,41,42,43,44]. In addition to plant-based compounds, probiotics and prebiotics such as Protexin and Naringenin have shown promise for *N. ceranae* management [45,46,47]. Synthetic compounds from several chemical classes including aminoglycosides, imidazoles, nitroimidazole, benzimidazoles, and benzodioxoles have also been tested on *Nosema* [32,46].

To expedite the search for new treatments to control *Nosema*, our research focused on testing compounds currently available on the market that are used to control endo-parasites in insects and other animals. The compounds chosen for this experiment include curcumin, ornidazole, fenbendazole, and nitrofurazone. Curcumin is a plant-based compound found in turmeric root *Curcuma longa* L., which was selected for its antimicrobial properties. Curcumin displays antimicrobial properties against bacteria, viruses, malaria, molds, yeast, and other fungi [48,49,50]. Strachecka et al. [39] showed that honey bees that consumed curcumin lived longer and had lower *Nosema* spore infections. However, tetrahydrocurcumin, a metabolite of curcumin, when evaluated against *N. ceranae*, had no significant reduction in spore load [46]. Ornidazole, a derivative of the antibiotic nitroimidazole, has been used against protozoa and bacteria [51,52,53,54]. It was chosen for this experiment because it demonstrated an 80–90% disease reduction in *Nosema bombycis* Nageli when administered orally to silk worm larvae, *Bombyx mori* L. [55]. However, in one study using cultured *N. ceranae*, ornidazole did not show activity against *N. ceranae* in vitro [32]. In other endo-parasites, Xin et al. [56] observed the potentially therapeutic inhibitory effects of ornidazole on glycolysis reactions (energy production) in *Echinococcus granulosus* and *Echinococcus multilocularis*, two hydatid worms in humans.

Other therapies for endo-parasitic organisms selected for this experiment include the anti-parasitic drug, fenbendazole. For six decades, benzimidazole derivatives (e.g., fenbendazole) have been used in agricultural, animal, and human medicines [57]. Activities documented against microsporidian species include the prophylactic oral administration of fenbendazole that effectively eliminated *Encephalitozoon cuniculi* (formerly *Nosema cuniculi*) infections in the central nervous system of rabbits [58,59]. Other animal studies found that fenbendazole effectively eliminated nematodes in cattle, turkeys and pigs [58,60,61,62,63,64,65]. For control of fish parasites, Park et al. [66] documented that fenbendazole at 30 mg L^−1^ significantly affected survival, growth and sex balance of the harlequin fly, *Chironomus riparius* Meigen, as well as disturbed the intracellular development of the microsporidia *Glugea anomala* Moniez [67]. Nitrofurazone, an antimicrobial organic compound belonging to the nitrofuran class, was chosen for this experiment due to its use in veterinary and human medicine. The antimicrobial properties of this compound make it effective against microsporidia, gram-positive and gram-negative bacteria [68,69]. Further, Ona et al. [70] documented that nitrofurazone acts as a direct inhibitor of DNA replication and damaged DNA in *Escherichia coli*.

The objectives of this project were to focus on in vivo screening of compounds available on the market to control *Nosema* disease in honey bees and to determine the sub-lethal effects of candidate compounds on honey bee longevity. The success of this research will provide beekeepers with promising alternative anti-fungal compounds to integrate into their *Nosema* management plan, while also reducing the risk of *Nosema* developing resistance to fumagillin. This ensures that beekeepers will have viable options to manage honey bee colonies responsibly and continue to maintain healthy bees.

## 2. Results

To evaluate the anti-*Nosema* activities of candidate compounds, we inoculated newly emerged house bees with *N. ceranae* spores at approximately 20 million spores per cage (approximately 0.2 million spores per bee (m.p.b)) at day zero. *Nosema* spores were below the level of microscopic detection in pre-experiment (day zero) samples of newly emerged bees. All caged bees were killed by day 7 when fed dimethoate as a cytotoxic agent in the death control group, therefore dimethoate values were excluded from the analyses. Molecular analyses confirmed the presence of *N. ceranae* and absence of *N. apis* in the inoculum.

### 2.1. Mean Abundance of Nosema Spores in the Mid-Experiment Sample

In live bees collected at days 5, 10 and 15 post-*Nosema* inoculation (p.n.i), the abundance of *Nosema* spores increased after day 5. The cumulative abundance was similar for the negative control (*Nosema*-inoculation free group) and fenbendazole (0.13 ± 0.58 m.p.b), where abundance of spores was lower than other treatments (over all days: F = 17.29; df = 6, 64; *p* < 0.0001) (Figure 1). The results indicated that the proliferation of spores at day 15 was lower in live bees treated with fenbendazole (0.36 ± 0.98 m.p.b). Curcumin (6.15 ± 0.8 m.p.b), nitrofurazone (3.2 ± 0.93 m.p.b), Fumagilin-B^®^ (3.21 ± 2.19 m.p.b), and ornidazole (2.89 ± 0.78 m.p.b) did not appear to have an inhibitory effect on spore proliferation in live mid-experiment bee samples when compared to fenbendazole (day 15: F = 22.19; df = 6, 61; *p* < 0.0001) (Figure 1).

### 2.2. Mean Abundance of Nosema Spores in the End-Experiment Samples

At the end of the experiment (day 20), all remaining live bees were collected to quantify *Nosema* spore abundance. Abundance of spores in replicates of the positive control reached 31.13 m.p.b (average 14.72 ± 2.32 m.p.b) without receiving medication over the 20-day trial. The lowest *Nosema* spore level (0.03 m.p.b) was counted in some replicates of the negative control (average 0.73 ± 3.27 m.p.b), Fumagilin-B^®^ (average 5.57 ± 3.17 m.p.b), and fenbendazole (average 0.36 ± 1.18 m.p.b). Based on the results, we found that the negative control and fenbendazole had similar, and lower, cumulative abundance of *Nosema* spores in the remaining live bees on day 20 p.n.i, compared to other treatments (F = 55.76; df = 6, 97; *p* < 0.0001) (Figure 2).

Differences were observed in the mean abundance of *Nosema* spores in live bees at the end-experiment samples, when different dilutions of each compound were compared to Fumagilin-B^®^: curcumin (F = 5.67; df = 5, 18; *p* = 0.0026); fenbendazole (F = 20.44; df = 5, 20; *p* < 0.0001); nitrofurazone (F = 1.59; df = 5, 19; *p* = 0.0211); and ornidazole (F = 3.2; df = 5, 23; *p* = 0.0243) (Table 1). The outcomes show that some dilutions of curcumin (10 mg L^−1^), nitrofurazone (≥100 mg L^−1^), and ornidazole (10 mg L^−1^ and 1000 mg L^−1^) had inhibitory effects on *Nosema* spore proliferation similar to the reference control (Fumagilin-B^®^). All dilutions of fenbendazole had a greater effect on the suppression of *Nosema* spore growth in treated bees, compared to the reference control.

### 2.3. Mean Abundance of Nosema Spores in the Dead Bee Samples

Dead bees were collected from the bottom of cages daily. The cumulative mean abundance of *Nosema* spores in dead bees increased during the trial. The analysis indicated that fenbendazole (0.03 ± 0.25 m.p.b) had the lowest cumulative abundance of spores in dead bees over time compared to other candidate compounds: ornidazole (0.63 ± 0.24 m.p.b), curcumin (0.74 ± 0.2 m.p.b), negative control (0.77 ± 0.19 m.p.b), nitrofurazone (1.78 ± 0.22 m.p.b), Fumagilin-B^®^ (3.5 ± 0.6 m.p.b), and positive control (4.95 ± 0.62 m.p.b) (F = 16.79; df = 6, 62; *p* < 0.0001) (Figure 3). The abundance of *Nosema* spores in dead bees increased after day 12 p.n.i. Dead bees in cages treated with Fumagilin-B^®^ had the highest mean abundance of *Nosema* spores after 20 days p.n.i (18.9 m.p.b).

### 2.4. Bee Mortality and Survival Rates

We found higher cumulative bee mortality in the positive control followed by curcumin and nitrofurazone when compared to other treatments. Analysis indicated that fenbendazole and ornidazole played a similar role in reducing bee mortality in *Nosema*-inoculated bees similar to Fumagilin-B^®^ and the negative control (F = 2.47; df = 6, 62; *p* = 0.0331) (Figure 4).

The percentage of bee mortality varied among dilutions for curcumin (F = 3.41; df = 5, 12; *p* = 0.038), nitrofurazone (F = 40.12; df = 5, 12; *p* < 0.0001), and ornidazole (F = 5.84; df = 5, 12; *p* = 0.0088), when compared to Fumagilin-B^®^ (Table 2). Although dilutions of fenbendazole (F = 0.53; df = 5, 12; *p* = 0.7532) had similar rates of bee mortality compared to Fumagilin-B^®^, some dilutions of curcumin (1000 mg L^−1^), nitrofurazone (≥100 mg L^−1^), and ornidazole (1000 mg L^−1^) killed more bees than Fumagilin-B^®^.

The Kaplan-Meier survival analysis identified differences among treatments with respect to bee mortality. Fenbendazole (71%) and Fumagilin-B^®^ (74%), along with the negative control (75%), had greater cumulative bee survivorship over 20 days p.n.i compared to other treatments (Long-Rank: *X*^2^ = 441.31; df = 6; *p* < 0.0001; Wilcoxon: *X*^2^ = 365.21; df = 6; *p* < 0.0001) (Figure 5). The survival rate for ornidazole (69%), nitrofurazone (57%), curcumin (44%), and the positive control (20%) was lower than the reference control. High mortality in *Nosema*-inoculated bees in the positive control indicated that *Nosema* inoculation at approximately 0.2 m.p.b reduced the bee population by more than 80% in the absence of a chemical control treatment.

A difference in survival rates of *Nosema*-infected bees fed different dilutions of fenbendazole (Long-Rank: *X*^2^ = 36.26; df = 5; *p* < 0.0001; Wilcoxon: *X*^2^ = 33.4; df = 5; *p* < 0.0001), curcumin (Long-Rank: *X*^2^ = 276.48; df = 5; *p* < 0.0001; Wilcoxon: *X*^2^ = 211.09; df = 5; *p* < 0.0001), nitrofurazone (Long-Rank: *X*^2^ = 1293.47; df = 5; *p* < 0.0001; Wilcoxon: *X*^2^ = 1119.86; df = 5; *p* < 0.0001), and ornidazole (Long-Rank: *X*^2^ = 126.58; df = 5; *p* < 0.0001; Wilcoxon: *X*^2^ = 107.23; df = 5; *p* < 0.0001) were observed in a non-dose-dependent manner. Non-significant survivorship was observed for dilutions of fenbendazole (≥1 mg L^−1^), curcumin (100 mg L^−1^), nitrofurazone (≤10 mg L^−1^) and ornidazole (≤100 mg L^−1^) compared to Fumagilin-B^®^ (Figure 6).

## 3. Discussion

In this investigation, we evaluate the antifungal activities of four compounds on their ability to inhibit *Nosema* spore replication and reduce the adverse effects of *Nosema*-infection on honey bee longevity. The compounds were chosen because of their reported efficacy against microsporidian species in bees and other animals. Our experiment inoculated bees with *N. ceranae* and determined the efficacy of each treatment over a 20-day period. Results revealed that fenbendazole showed the most promise among tested compounds, followed by ornidazole as an anti-*Nosema* agent. Fenbendazole reduced the number of *N. ceranae* spores by 91%, compared to bees in the positive control group. This active ingredient had no negative effect on honey bee longevity; in fact, it improved bee survival compared to the *Nosema*-infected non-treated control group. Ornidazole did not inhibit *Nosema* spore growth in bees, compared to fenbendazole, but bee longevity was improved when *Nosema*-infected bees were fed rates ≤100 mg L^−1^.

To determine *Nosema* spore abundance post-inoculation, live and dead bees were collected throughout the experiment. Spore abundance analyzed mid-experiment using live honey bee samples demonstrated that honey bees can be infected with *Nosema* using a mass feeding inoculation, with similar results to those normally found with individual feeding [18,22,71,72]. This feeding method closely mimics trophallaxis, which is the most common way that *Nosema* infection spreads within a colony. Mass feeding also replicates the typical method for administering *Nosema* treatments to a bee colony, such as Fumagilin-B^®^. In our investigation, newly-emerged house bees were inoculated with approximately 0.2 million *Nosema* spores per bee at day zero using the mass feeding method. Live bees were then sampled every five days to track *Nosema* spore abundance over time. We counted and detected the presence of *Nosema* spores on day 5 p.n.i using microscopy. We did not test live bees before this time, but when Glavinic et al. [26] tested bees on day 3 p.n.i, no spores were detected in live bee samples. In other experiments, *Nosema* detection dates varied, with spores not observed until day 6 and 12 p.n.i [18,22,73,74,75]. This suggests that time is required to establish an extensive infection in the gut epithelial cells of honey bees [75]. Discrepancies in spore detection times could be associated with initial spore abundance, bee stressors, inoculation, and incubation methods.

In our experiment, spore abundance increased in all inoculated treatments throughout the entire length of the trial in both live and dead bees. In live bees, the highest abundance of spores was detected in the positive control and in bees treated with curcumin. Interestingly, the mean *Nosema* abundance in the positive control peaked at day 10 and decreased at day 15. This is likely due to the increase in bee mortality of heavily infected bees after day 10, leaving the remaining live bees sampled at the end of the trial with lower *Nosema* infections. Fenbendazole had consistently lower mean *Nosema* spore abundance in both live and dead bees collected throughout the experiment, highlighting its ability to control *Nosema* proliferation in honey bees. Mean spore abundance in bees treated with Fumagilin-B^®^ was similar to nitrofurazone and ornidazole in live bees; however, spore abundance in dead bees treated with Fumagilin-B^®^ was similar to dead bees from the positive control. An explanation for this might include the impact of fumagillin and/or the fumagillin–*Nosema* synergistic effects on bee longevity. Some studies reported that despite fumagillin being considered an effective antibiotic in *Nosema* management [14,15], the dicyclohexylamine salt in the compound induced an immune-suppression response in bees [26] and a reduction in bee longevity [22]. Glavinic et al. [26] found that fumagillin caused bee mortality in *Nosema*-free bees similar to *Nosema*-infected groups. On the contrary, Braglia et al. [76] found that infected bees treated with fumagillin had low bee mortality and was effective at controlling *Nosema*. The impact of fumagillin on *Nosema-*infected bees needs to be explored further, as the results of our study may indicate that Fumagilin-B^®^ at the recommended dose, in addition to suppressing spore production, increased the mortality of *Nosema*-infected bees.

All live bees remaining at the end of the experiment were analyzed to quantify *Nosema* infection levels at twenty days after inoculation. At the end of the trial, curcumin did not show promising results as an effective control of *Nosema* spores. Extractions of curcumin found in turmeric plants have been shown to possess a variety of suppression effects on parasites and pathogens [77,78,79,80]. A number of studies have documented the beneficial effects of curcumin or curcumin metabolites on honey bees and *Nosema* spp. [39,48,81]. Di Pasquale et al. [81] indicated that tetrahydrocurcumin, a metabolite of curcumin, increased immune responses in honey bees. Strachecka et al. [39] fed *Nosema*-infected bees with 3 mg L^−1^ of curcumin and found that worker bees had higher dilutions of hemolymph proteins, an increase in activity of antioxidant enzymes, higher total antioxidant potential, and in addition, the activities of neutral proteases and their inhibitors were boosted. Recently, Borges et al. [46] tested tetrahydrocurcumin on *Nosema*-infected bees. They reported that this compound was able to reduce spore counts and increase the survival and health of *Nosema*-infected bees. Our results do not support these findings as the abundance of *Nosema* spores was not inhibited in curcumin-treated bees. Bees were fed a broad dilution of curcumin (≤1000 mg L^−1^), and only the 100 mg L^−1^ dose showed a survival rate similar to Fumagilin-B^®^. In fact, curcumin-fed bees had the highest cumulative mortality out of all candidate compounds evaluated. This could be due to either toxic effects from chronic exposure to curcumin, or from the *Nosema*-chemical interaction. Due to our results, curcumin will not be considered in future tests at the colony level.

Similar to curcumin, nitrofurazone did not show promising results as an effective control of *N. ceranae*. Nitrofurazone is in the chemical class nitrofuran. It has been used for human health [82], and as an anti-parasite treatment in fish [83] and poultry [84]. We found nitrofurazone to be highly toxic, when *N. ceranae*-infected bees received a dose of ≥100 mg L^−1^. Moffett et al. [85] reported a high mortality in *N. apis*-infected bees fed with 200 mg L^−1^ nitrofurazone. In our study, lower dosages (≤10 mg L^−1^) were safe for bees; however, nitrofurazone did not adequately reduce *Nosema* spore levels (>9 m.p.b) in the end-experiment samples. As such, this candidate compound will not be considered for further investigations.

Ornidazole, a synthetic nitroimidazole, was chosen for this study due to promising reports for controlling endo-parasites [86,87,88]. Although research on ornidazole for the control of *Nosema* in vivo on honey bees is limited, research has shown that oral application prevented *Nosema bombycis* in silkworm larvae [55]. In cell cultures, Gisder and Genersch [32] used ornidazole against *N. ceranae*. They found that it was highly toxic for cells (89% death rate), but did not show any significant activity against *N. ceranae*. In our study, ornidazole inhibited *Nosema* spore proliferation as the dosage increased; however, at a dosage of 1000 mg L^−1^ it was found to be toxic to honey bees. Interestingly, at a dosage of ≤100 mg L^−1^, ornidazole improved the longevity of infected bees. Overall, this compound did not play an important role in *Nosema* control compared to fenbendazole, but showed promise at certain dilutions. This product also had the potential to improve the longevity of *Nosema*-inoculated bees at lower doses, similar to Fumagilin-B^®^. This suggests that ornidazole at certain doses might be able to inhibit *Nosema* spore proliferation in infected bees. Based on these findings, future investigation is required to elucidate the effect of ornidazole on *Nosema* and honey bee development.

To the best of our knowledge, the effects of fenbendazole on *Nosema* spp. or *A. mellifera* have not been evaluated, and thus our study is the first evaluation. Fenbendazole is a benzimidazole derivative that is considered an anti-endoparasitic agent used in veterinary medicine against gastrointestinal parasites in fish and domestic animals [89,90]. Although the mode of action for fenbendazole against *Nosema* spp. has not been determined, it has been shown to cause a decline in intracellular proliferation of *Cryptococcus* spp., a fungal pathogen in humans. The anti-fungal activity of fenbendazole has been associated with microtubule disorganization and reduced phagocytic escape through vomocytosis [57,91]. Benzimidazoles attack the protein β-tubulin in microorganisms, suppress the microtubule polymerization action, and cause depolymerization and cell death in the organism [92]. Although this anti-fungal has been successful at controlling a variety of organisms, benzimidazole resistance has been reported in nematodes and fungi [93,94,95,96]. The mutation in benzimidazole-resistant helminths and fungi has been linked to mutations in β-tubulin genes [97], and were reported in *Haemonchus, Trichostrongyfus* spp., *Encephalitozoon* spp., *Enterocytozoon* spp. and *Vittaforma* spp. (former *Nosema* spp.) [98,99]. Understanding the mode of action of fenbendazole on *N. ceranae* could be important to limit resistance development in *Nosema,* if it is to be used as a control option.

It is worth noting that the active ingredient of fenbendazole at a rate of ≤1000 mg L^−1^ had a deleterious effect on *N. ceranae* in *A. mellifera*. Although all dilutions reduced the abundance of *Nosema* in bees, the bee mortality rate in our experiment was not dose-dependent. The bee mortality rate was high (>31%) at certain low and high dilutions of fenbendazole, but overall mortality was similar to the Fumagilin-B^®^ treatment group. This indicates that at higher doses, fenbendazole may be toxic to bees, but also that bee mortality at lower dilutions may be more likely associated with other stressors (e.g., chronic parasitization by *N. ceranae*) and not with *Nosema*-chemical interactions. Various stressors including antagonistic, additive, or synergistic effects may have affected bee longevity in our trials. Despite the stress factors to which bees were exposed in this study, the Kaplan-Meier survival analysis showed similarities in survival rates among the negative control, fenbendazole, and Fumagilin-B^®^ treatment groups. This indicates that aside from observing bee mortality at low dilutions, fenbendazole could play a role in reducing stress pressures on bees. We suggest adjusting the effective and bee safe dosage by determining the minimum inhibitory concentration (MIC) of fenbendazole on *Nosema* in honey bee colonies. Once established, the opportunity exists to further test fenbendazole against *Nosema* infections at the honey bee colony level in field trials and to determine the risk of detecting residue in honey and other bee products.

For future evaluations, it is important to explore the exposure of bees to stressors, which affect the hosts’ physiology and immune system [16,19]. Goblirsch et al. [17] showed that placing bees in cages with no queen and or brood pheromone may affect the behavioral development of bees. In this bioassay, confined bees experienced stressors such as the absence of a queen, lack of queen and brood pheromones, nutritional stress, lack of opportunity for flight or defecation, chemical exposure, and incubation. In this study, if stress factors played a role in spore abundance, it did not appear to affect bees equally, across compound groups. For the curcumin treatment group, it appeared that the bees could not cope with potential stressors and that the chemical may have encouraged *Nosema* spores to proliferate faster in bees, compared to other treatments. Another possible reason for the increase in spore abundance could be changes in compound properties caused by the solvent (sugar syrup and acetone) used, and/or environmental elements such as light, temperature, and humidity. Research has shown that ambient light and sunlight affect the stability of fumagillin, which if exposed is less effective at treating *Nosema* [23,100]. However, all replicates (cages) were treated in the same manner and environmental conditions to minimize the variability among results. Since the cages were kept in a dark environment (incubator), it was assumed that light was not a major factor affecting the compounds’ ability to inhibit *Nosema* spore proliferation. However, possible abiotic environmental factors and solvent reactions that affect the bioactivity of agents and the stability of active ingredients should be considered in future laboratory bioassay cage and field trials.

In addition, to advance this research, the evaluation of winter bees will be necessary to determine the success of these compounds. Winter bees are reared in late summer on the Canadian prairies [101], and are adapted to live longer in a confined colony during the long, cold winters of Canada’s northern climate. A vast difference in protein content [102,103], antioxidative enzymes, antimicrobial peptides [104], and gut microbiota [105] have been documented in summer vs. winter bees. The treatment differences between summer and winter bees is highlighted by Braglia et al. [76], who found a lower abundance of *N. ceranae* in winter bees, compared to summer, when bees were fed acetic acid at high concentrations (0.35 M per 1 mL syrup (1:1 *w:v*)). In addition, Bernklau et al. [106] showed that *p*-coumaric acid could reduce the development of *Nosema*, but only on winter honey bees. For fenbendazole and ornidazole, more studies are required to assess their efficacy on *N. ceranae* and *N. apis* on mixed-age winter bees under laboratory and field conditions.

Although fenbendazole has shown promising results in controlling *Nosema* and preventing honey bee mortality, it is important to consider the implication of in-hive applications and whether this substance is toxic to humans. Fumagillin, for instance, has been used in North America for decades; however, it has been banned in other countries due to the potential toxicity to humans [32,33,34]. Fumagillin and its analogs have historically been used to treat tumors in humans [107,108]. Toxicological studies determined that fumagillin (LD_50_ = 2000 mg/kg) is more lethal than fenbendazole (LD_50_ ≥ 10,000 mg/kg) in mice when administered orally [61,109]. This indicates that the potential risk of fenbendazole to human health may be lower in comparison to fumagillin. Fenbendazole is not listed as a carcinogenic agent. However, Villar et al. [110] showed that fenbendazole may promote tumors in rats. This indicates that there is a need to assess the risks to human health and determine the residues in bee products. Although promising, fenbendazole needs to be assessed before being made commercially available to producers.

## 4. Materials and Methods

European honey bee (*A. mellifera*) colonies (n = 6) headed by Kona queens (Hawaii, USA) were housed in Langstroth double brood-chambered boxes, and managed using standard management practices to minimize variabilities among tested honey bee populations [111]. Alberta best management practices were applied to control *Varroa* mites, *Varroa destructor* Anderson and Trueman, using Apivar (500 mg of amitraz/strip; Veto Pharma, Palaiseau, France), or oxalic acid (Medivet Pharmaceutical Ltd., High River, AB, Canada), according to manufacturers’ recommendations [112]. Colonies selected for this experiment did not receive fumagillin in the fall or spring to prevent experimental contamination. In autumn, the colonies were fed sugar syrup and overwintered outdoors [113]. This laboratory experiment was carried out at the Crop Diversification Centre North, Edmonton, Alberta, Canada (53°38′32.9″ N 113°21′47.1″ W) in summer 2018.

In this study, a Plexiglas cage (11 × 12 × 15 cm; outer dimensions; henceforth referred to as ‘cage’) was designed specifically for this bioassay experiment (Figure 7). Inside the cage, a piece of plastic foundation (4 × 5 cm) was attached to the middle of the roof between feeding holes to allow the bees to cluster. The bottom of the cage had two removable mesh screens (10 × 11 cm) to make easier the removal of dead bees for *Nosema* spore analysis. The walls of the cage had holes for air ventilation. The top of the cage was designed to hold four 15 mL centrifuge gravity feeder tubes (VWR, Canada). The gravity feeders were created by puncturing holes in the lid of the 15 mL tube and inverting them. Each tube contained either water or sugar syrup (1:1, *w*/*w*). A tray (1.5 × 9.5 × 10.5 cm) was inserted at the bottom of each cage. Each cage had two sliding side walls (10 × 15.5 cm) to introduce or remove bees from the cage.

### 4.1. Isolation of Nosema Spores to Be Used for Inoculation

*Nosema* spores were collected from fecal material on the hive’s body from highly infected colonies and mixed with water to make high density of inoculum. Mixed-age bees from colonies were placed in wooden cages (8.5 × 11.5 × 15 cm) in the laboratory and fed 10 mL 50% sugar solution using gravity feeder tubes containing high levels of *Nosema* spores. Each cage was incubated at 33 ± 2 °C and 60 ± 5% relative humidity (RH) for up to 15 days, allowing the *Nosema* spores to reproduce in the intestinal tract of the honey bee [18]. After 15 days the cages were placed in the freezer (−20 °C) until 100% bee mortality. To obtain high concentrations of fresh *Nosema* spores, the ventriculi were removed from the dead honey bee abdomen and homogenized with a mortar and pestle in water. Spores were counted using a light microscope (×400) and hemocytometer (Neubauer improved bright-light, 0.0025 mm^2^) to determine total spores per mL [114].

### 4.2. Newly Emerged Bee Collection and Caging

In order to collect relatively disease-free newly-emerged bees, frames of sealed worker brood were removed from healthy fumagillin-free colonies with no or low *Varroa* mite levels (<1%), and individually confined in wooden brood emergence cages (50 × 26 × 7 cm). All adult bees were removed from the brood frames to avoid contamination. Each emergence cage had a screen on both sides for ventilation and was kept in an incubator at 33 ± 2 °C with 60 ± 5% RH until bees emerged from their cells. Once emerged, the bees were collected in a plastic Rubbermaid container and exposed to carbon dioxide until anesthetized. Approximately 100–120 bees (12 ± 1 g) were placed in each cage. At the same time, a sample of 50 newly emerged bees was collected to estimate the pre-experiment *Nosema* levels. Drone bees were excluded from the experiment.

### 4.3. Experimental Procedures

The biological activities of four antimicrobials including curcumin ((E,E)-1,7-bis(4-Hydroxy-3-methoxyphenyl)-1,6-heptadiene-3,5-dione), fenbendazole (methyl-5-(phenylthio)-2-benzimidazolcarbamate), nitrofurazone (5-Nitro-2-furaldehyde semicarbazone), and ornidazole (1-(3-Chloro-2-hydroxypropyl)-2-methyl-5-nitroimidazole) were evaluated against *N. ceranae*. All active ingredients were provided from Sigma-Aldrich, Canada. The commercially available product Fumagilin-B^®^ (Medivet, Pharmaceuticals, Alberta, Canada) was used as a reference control treatment. Positive (*Nosema* spores only), negative (no *Nosema* or chemical), and death (dimethoate) controls were also included in the experiment. Dimethoate (Sigma-Aldrich, Oakville, ON, Canada), an organophosphate insecticide, was used as a death control toxin to validate bioassay outputs [115]. Treatments were tested with three replicates (cages) for each dilution. Subsequent serial dilutions for each candidate compound were prepared using a 50% sugar solution in water (*w*/*w*) to provide the following dilutions: 1000 mg L^−1^, 100 mg L^−1^, 10 mg L^−1^, 1 mg L^−1^, and 0.1 mg L^−1^. A Fumagilin-B^®^ dilution was made at 25 mg L^−1^ based on manufacture’s recommendations (Medivet Pharmaceuticals, High River, AB, Canada). To increase solubility of active ingredients, they were first dissolved in 10 mL acetone (≥99.5%; density 784 g mL^−1^; Fisherbrand, Fisher Scientific, Ottawa, ON, Canada), then mixed with 50% sugar solution to make stock solutions. The stock solutions were prepared in new 50 mL centrifuge tubes and agitated on a vortex mixer (VWR, Mississauga, ON, Canada) for a period of 2–3 min for same-day use. Acetone was added into the syrup used for stock solutions in positive, negative, death, and reference controls. The aqueous stock solutions were clear. All interactions with chemicals were conducted under the fume hood with operators wearing a full-face respirator mask (6900, 3M, Saint Paul, MN, USA) including filters (60923, 3M, Saint Paul, MN, USA) and other PPE.

On day zero of the experiment, approximately 100–120 (12 ± 1 g) bees were placed in the cages and starved for a 2 h period. After 2 h, all dead bees were removed using six-inch tweezers and discarded. Two 15 mL gravity feeder tubes were inverted on the top of each cage, one containing water and the other containing sugar solution specific to the applied treatment group. Positive control, death control, and candidate chemical groups (including reference control, Fumagilin-B^®^) were mass-inoculated and fed 50% sugar solution containing fresh *Nosema* spores (20 million spores per cage), and the negative control group was fed 50% sugar solution only. All cages were placed in the incubator overnight. The gravity feeders containing *Nosema* spores remained on the cages until the solution was consumed.

On day one of the experiment, the treatment groups, including the Fumagilin-B^®^, received feeder tubes containing the candidate compounds in 50% sugar solution. Positive and negative controls only received sugar syrup. Death control cages were fed diluted Dimethoate Pestanal^®^ 0.0033 mg per cage in sugar solution [116]. For the remainder of the experiment, all treatment cages were provisioned *ad libitum* with 50% sugar solution containing a candidate compound, Fumagilin-B^®^, or dimethoate for 20 days. Positive and negative control groups were provisioned *ad libitum* with a sugar solution for the duration of the experiment (Figure 8). All bioassay cages were randomly placed in a controlled-environment incubator at 33 ± 2 °C with 60 ± 5% RH in the dark for 20 days p.n.i. Temperature (°C) and relative humidity (%) in the incubator were recorded using HOBO data loggers (Onset Computer Corporation, Bourne, MA, USA).

Dead bees were collected from the bottom of cages daily (days 1–20) and placed in 15 mL centrifuge tubes (Fisher Scientific, Canada) to determine daily *Nosema* infection levels. On days 5, 10, and 15 p.n.i, five live bees were randomly collected in 15 mL centrifuge tubes as “mid-experiment” samples from each cage. At the end of the trial on day 20, all remaining live bees were placed in a freezer (−20 °C) and then collected in 50 mL centrifuge tubes (Fisher Scientific, Canada) as “end-experiment” samples (Figure 2). All dead and live bee samples were stored in a freezer before processing. To determine the *Nosema* spore abundance (m.p.b), bees were processed in composite samples with a ratio of 1 mL of water per bee. Once the samples were processed, *Nosema* spores from each experiment were counted using a hemocytometer and microscopic method [114].

To avoid additional *Nosema* spore contamination or pesticide residues from commercial pollen patties, the bees were caged without a pollen protein source. To prevent cross-contamination at the end of the experiment, each cage was rinsed with ethanol (90%) to remove any chemical residues, and then triple-washed by hand using detergent and placed in an industrial dishwasher for one rinse cycle. In addition, the mesh on the top of the cages holding the feeders was replaced. All experimental components in contact with chemicals (e.g., stock solution tubes, feeder tubes and metal screens) were disposed of appropriately.

### 4.4. Nosema Species Assessment

*Nosema* species were determined using multiplex PCR as described by Copley et al. [117]. The DNA was extracted from freshly collected *Nosema* spores using the DNeasy Blood and Tissue Kit (Qiagen, Valencia, CA, USA) following the manufacturer’s protocols. The DNA quantity and quality was determined using a NanoDrop Spectrophotometer (Thermo Scientific, Wilmington, DE, USA). In this study, primers 218MitocF (5′-CGGCGACGATGTGATATGAAAATATTAA-3′) and 218MitocR (5′-CCCGGTCATTCTCAAACAAAAAACCG-3′) for *N. ceranae*, and 321ApisF (5′-GGGGGCATGTCTTTGACGTACTATGTA-3′) and 321ApisR (5′-GGGGGGCGTTTAAAATGTGAAACAACTATG-3′) for *N. apis* (Invitrogen, Burlington, ON, Canada) were used. The thermal cycling was one cycle at 95 °C for 10 min (initial denaturation), followed by 35 cycles at 95 °C for 30 s (denaturation), 60 °C for 30 s (annealing), 72 °C for 30 s (extension), followed by one cycle of at 72 °C for 7 min (final extension), using the Thermal PCR System (Bio-Rad, Mississauga, ON, Canada). All mixed primers were tested with negative and positive controls. DNA was then run through 1% agarose gel electrophoresis (Bio-Rad, Mississauga, ON, Canada) to confirm species presence or absence.

### 4.5. Statistical Analyses

The bioassay experimental design was a split-plot treatment arrangement in a randomized complete block design with treatments consisting of four candidate compounds, one reference control, one positive control, and one negative control. Fumagilin-B^®^ had one dilution, and each candidate compound had five dilutions, with three replicates for each dilution. For this experiment, treatments were the main plots and dilutions were the sub-plots. The effect of treatments on bee longevity was analyzed using the Kaplan-Meier survivorship analysis, (PROC LIFETEST) [118]. Prior to analyses, proportions for bee mortality rate and mean abundance of *Nosema* spores were arcsine transformed and log transformed, respectively [119]. All data are presented as untransformed means. The effects of treatments on mean abundance of *Nosema* spores in mid-experiment bees (sampled on days 5, 10, and 15) and in dead bees were analyzed by ANOVA using a repeated measures analysis of variance with an autoregressive heterogeneous covariance structure. Treatments were analyzed as main effects, sampling day as repeated measure, and cage as subject (experimental unit), (PROC MIXED) [118]. Differences among treatment means were compared using the Bonferroni correction [118].

## 5. Conclusions

In this study, for the first time, the biological activity of the active ingredient fenbendazole was tested on *N. ceranae* in honey bees. Our findings demonstrate that among tested compounds evaluated, fenbendazole showed the most promise for controlling *N*. *ceranae* infections in honey bees under laboratory conditions. Fenbendazole and ornidazole both improved bee longevity compared to the reference control, Fumagilin-B^®^. Other candidate compounds, including curcumin and nitrofurazone, did not have an effective inhibitory action against *Nosema* spores and both treatments resulted in lower bee longevity. Fumagilin-B^®^, the standard control product, suppressed *Nosema* spores in the bioassay compared to the positive control. However, it also resulted in higher bee mortality. Although fumagillin has been an effective way to control *Nosema*, there is a risk of *Nosema* developing resistance to fumagillin [28]. Furthermore, fumagillin has been shown to cause immune suppression and oxidative stress [26] in bees. It is also toxic to bees and humans at certain levels [22] and leaves residues in bee products [120]. As such, investigating new alternative compounds to use against *Nosema* is necessary. Based on reported results, among the four pure synthetic compounds screened, fenbendazole and ornidazole had potential anti-nosemosis activities and increased the survival rate of *Nosema*-infected bees in our in vivo experiment. Further studies are required in the field to confirm these results; however, our findings offer a promising avenue for exploring new therapeutic agents to control *Nosema* infections in honey bees in the future.

## Figures and Tables

**Figure 1 pathogens-11-01375-f001:**
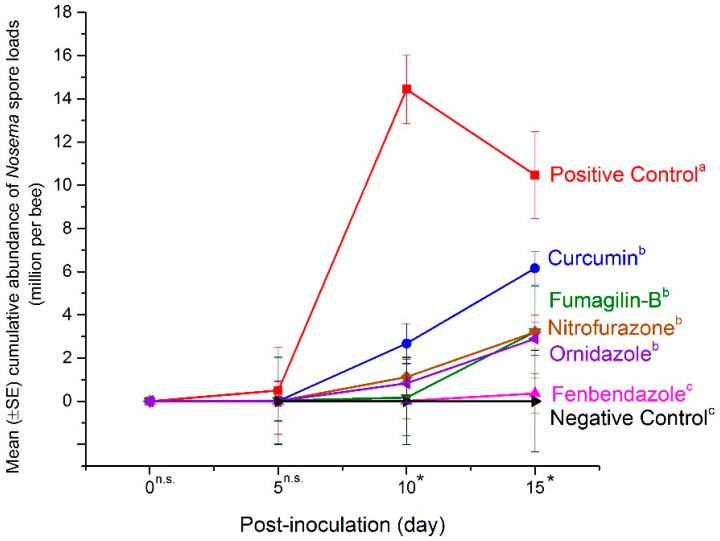
Cumulative effects of anti-fungal activities for tested compounds on mean (±SE) *Nosema* spore abundance (m.p.b) in mid-experiment live bees. Each point indicates the average abundance for replicates in each compound (n = 15; 5 dilutions × 3 replicates), Fumagilin-B^®^ (n = 3), negative (n = 3), or positive (n = 3) controls. Vertical bars on each point indicate the standard error (±SE). Asterisks and n.s. indicate significant and non-significant differences, respectively, within treatments for each time point of sampling. Means followed by different letters are significantly different among treatments on day 15 (*p <* 0.05).

**Figure 2 pathogens-11-01375-f002:**
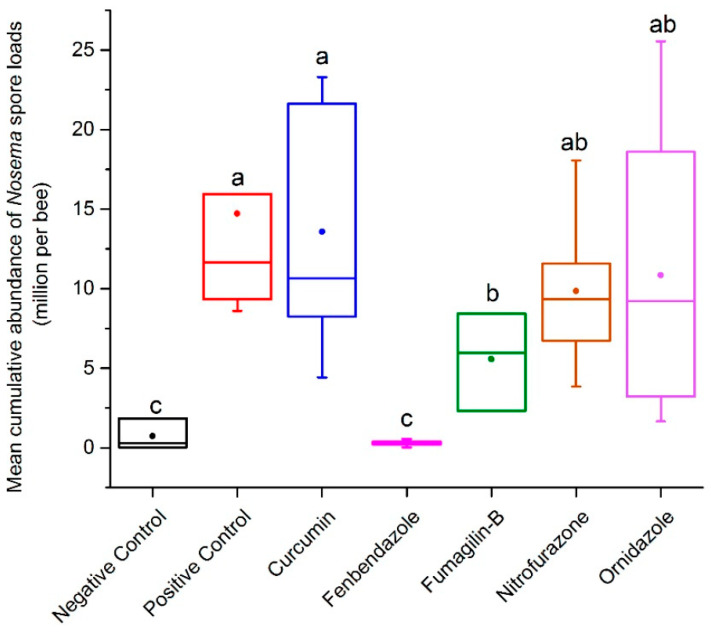
Cumulative effects of anti-fungal activities of tested compounds on mean *Nosema* spore abundance (m.p.b) in live bees at the end-experiment samples (day 20). Each box represents the average abundance for replicates in each compound (n = 15; 5 dilutions × 3 replicates), Fumagilin-B^®^ (n = 3), negative (n = 3), or positive (n = 3) controls. The boxplots indicate the standard error (length of box), mean (solid dot), median (horizontal line inside box), 5th and 95th percentiles (lower and upper vertical lines). Means followed by different letters are significantly different among treatments (*p <* 0.05).

**Figure 3 pathogens-11-01375-f003:**
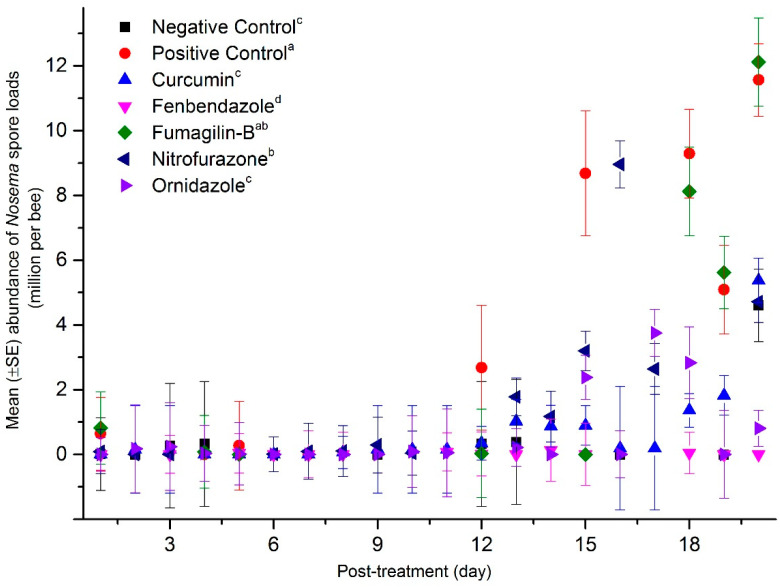
Cumulative effects of anti-fungal activities of compounds on mean (±SE) *Nosema* spore abundance (m.p.b) in dead bees collected daily during the experiment. Each symbol represents the average abundance for replicates in each compound (n = 15, 5 dilutions × 3 replicates), Fumagilin-B^®^ (n = 3), negative (n = 3), or positive (n = 3) controls for each time point of sampling. Vertical bars on each point indicate standard error (±SE). In the graph, treatments followed by different letters are significantly different over the experiment (cumulative effects) (*p <* 0.05).

**Figure 4 pathogens-11-01375-f004:**
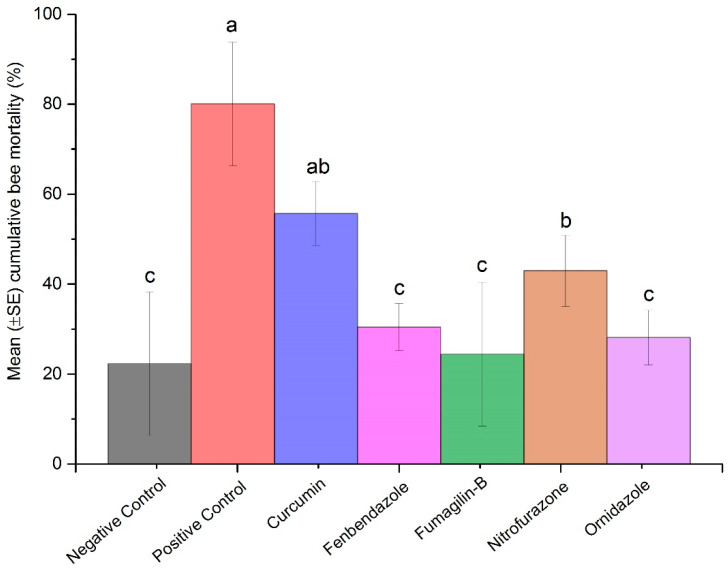
Mean (±SE) cumulative bee mortality (%) in tested compounds during 20 days of trial. Each bar represents the average cumulative bee mortality of replicates for each compound (n = 15, 5 dilutions × 3 replicates), Fumagilin-B^®^ (n = 3), negative (n = 3), or positive (n = 3) controls. Each vertical line indicates ± standard error (±SE). Means followed by different letters are significantly different among treatments (*p <* 0.05).

**Figure 5 pathogens-11-01375-f005:**
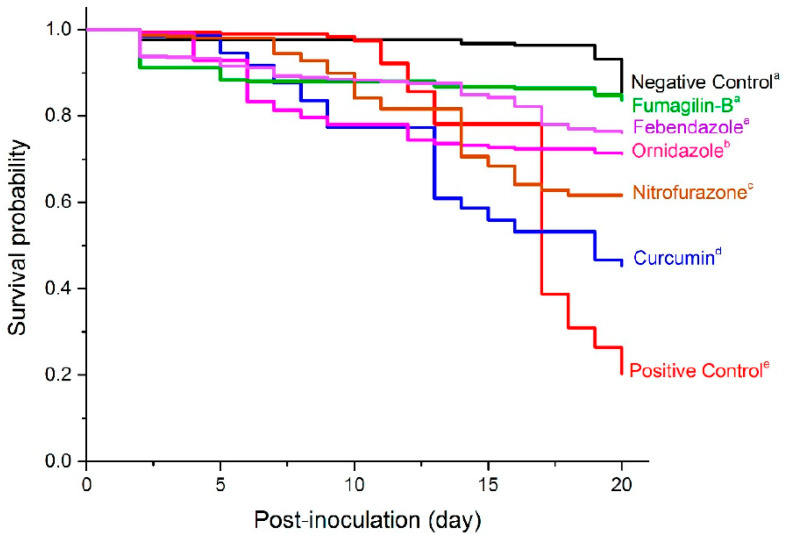
Kaplan-Meier cumulative effects of anti-fungal activities of tested compounds on bee survival probability in different treatments over 20 days p.n.i. Each line indicates the average survival rate for replicates in each compound (n = 15, 5 dilutions × 3 replicates), Fumagilin-B^®^ (n = 3), negative (n = 3), or positive (n = 3) controls. Means followed by different letters are significantly different among treatments (Wilcoxon multiple comparison, *p <* 0.05).

**Figure 6 pathogens-11-01375-f006:**
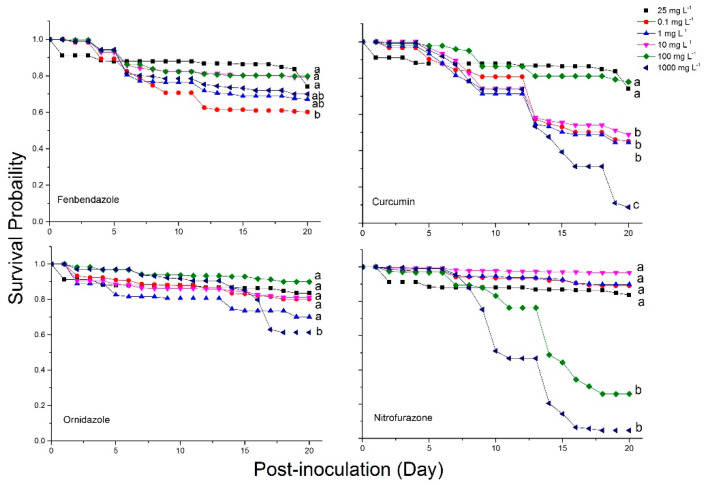
Kaplan-Meier anti-fungal activities of candidate compounds on the bee survival probability in different dilutions of compounds over 20 days p.n.i. Each point indicates the average survival rate for replicates in each dilution (n = 3). The dilution 25 mg L^−1^ (black square) represents the Fumagilin-B^®^ treatment. Means followed by different letters are significantly different among dilutions of each candidate compound and Fumagilin-B^®^ (Wilcoxon multiple comparison, *p <* 0.05).

**Figure 7 pathogens-11-01375-f007:**
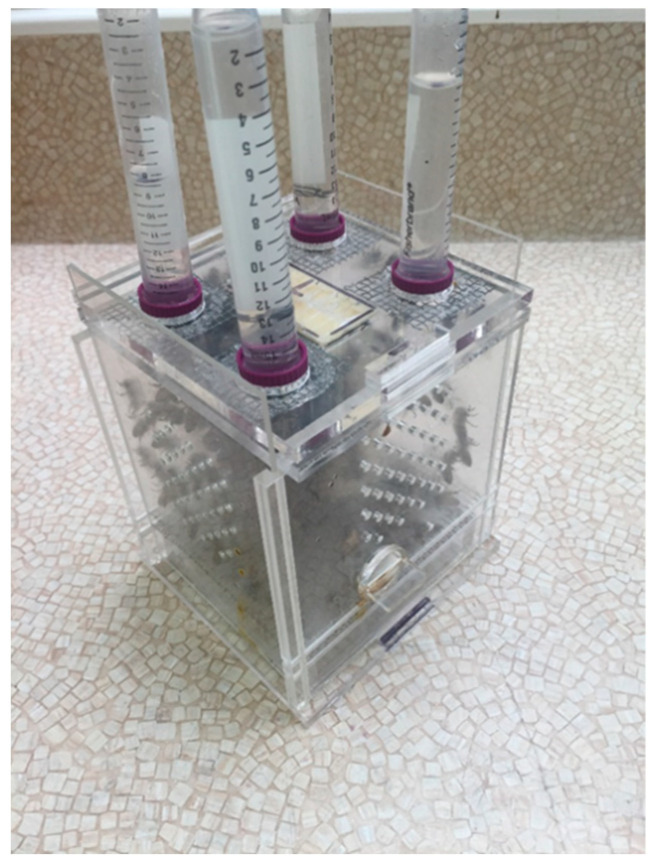
Plexiglas bioassay cage designed specifically for antibiotics screening bioassay experiment. Cage includes a piece of plastic foundation, two removable mesh screens, holes on side walls, gravity feeder tubes on top, a tray, and two sliding side walls.

**Figure 8 pathogens-11-01375-f008:**
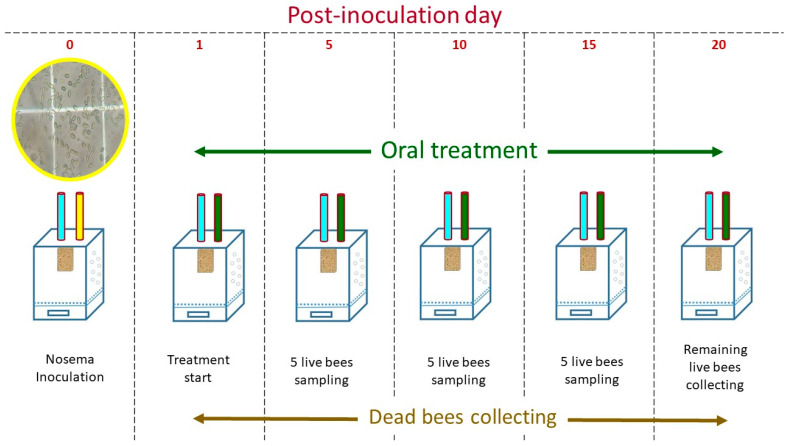
Experimental procedure: In this bioassay cage study, a group of newly emerged bees inoculated with freshly collected *Nosema* spores at day zero (yellow tube). Bees were fed *ad libitum* water (blue tube) and different dilutions of candidate compounds or Fumagilin-B^®^ (green tube) in 50% sugar syrup. Positive and negative control groups were provisioned *ad libitum* with sugar solution only over 20 days. Each day, dead bees were collected from the bottom of the cages. On days 5, 10, and 15 p.n.i, five live bees were randomly collected as “mid-experiment” samples from each cage. On day 20, all remaining live bees were collected as “end-experiment” samples from the cages.

**Table 1 pathogens-11-01375-t001:** Mean (±SE) *Nosema* spore abundance (m.p.b) for the 5 dilutions of each treatment in live bees for end-experiment samples (day 20) in comparison with Fumagilin-B^®^. Each value indicates averages for three replicates ± standard error (±SE). Means followed by different letters are significantly different within each row (*p <* 0.05).

Treatment	1000 mg L^−1^	100 mg L^−1^	10 mg L^−1^	1 mg L^−1^	0.1 mg L^−1^	Fumagilin-B^®^
Curcumin	13.75 ± 2.91 ^a^	16.44 ± 2.06 ^a^	5.55 ± 2.52 ^b^	13.73 ± 2.48 ^a^	12.02 ± 2.52 ^a^	5.57 ± 3.17 ^b^
Fenbendazole	0.25 ± 2.26 ^b^	0.42 ± 2.06 ^b^	0.4 ± 2.26 ^b^	0.43 ± 2.52 ^b^	0.29 ± 2.91 ^b^	5.57 ± 3.17 ^a^
Nitrofurazone	5.54 ± 5.05 ^bc^	7.38 ± 3.57 ^b^	9.28 ± 1.91 ^b^	11.94 ± 2.06 ^a^	9.98 ± 2.06 ^b^	5.57 ± 3.17 ^c^
Ornidazole	2.73 ± 2.52 ^b^	11.05 ± 2.1 ^a^	8.5 ± 2.06 ^b^	18.64 ± 2.52 ^a^	13.23 ± 2.1 ^a^	5.57 ± 3.17 ^b^

**Table 2 pathogens-11-01375-t002:** Mean (±SE) bee mortality (%) during the 20 day exposure to different dilutions of treatments in comparison with Fumagilin-B^®^. Each value indicates the average of three replicates ± standard error (±SE). Means followed by different letters are significantly different within each row (*p <* 0.05).

Treatment	1000 mg L^−1^	100 mg L^−1^	10 mg L^−1^	1 mg L^−1^	0.1 mg L^−1^	Fumagilin-B^®^
Curcumin	94.44 ± 9.35 ^a^	24.22 ± 9.55 ^b^	50.67 ± 9.5 ^ab^	53.69 ± 9.3 ^ab^	55.44 ± 9.45 ^ab^	24.43 ± 14.28 ^b^
Fenbendazole	31.22 ± 8.5 ^a^	21.23 ± 8.99 ^a^	20.92 ± 7.99 ^a^	34.79 ± 9.31 ^a^	44.06 ± 9.55 ^a^	24.43 ± 14.28 ^a^
Nitrofurazone	98.26 ± 9.25 ^a^	81.75 ± 9.62 ^a^	4.41 ± 8.77 ^c^	12.95 ± 8.9 ^bc^	17.4 ± 9.11 ^bc^	24.43 ± 14.28 ^b^
Ornidazole	56.35 ± 9.33 ^a^	12.41 ± 9.05 ^b^	22.16 ± 8.88 ^b^	23.95 ± 9.74 ^b^	25.68 ± 9.67 ^b^	24.43 ± 14.28 ^b^

## Data Availability

The data presented in this study are available on requested from the corresponding author.

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
