# Peer review of "In Vivo Inhibitory Assessment of Potential Antifungal Agents on Nosema ceranae Proliferation in Honey Bees"

_pathogens, 2022, doi:10.3390/pathogens11111375_

Round 1

Reviewer 1 Report

This manuscript is well-written and researched in the introduction and the abstract; however, things are getting worse in the following sections. The results are not impressive, the discussion is far worse than the introduction, and the M&Ms are problematic with some technique flaws. These findings made me wonder if the results reported in this manuscript are real facts. Flaws in M&Ms are not acceptable for papers that trying to identify drugs. Since the current understanding of research on alternative treatments for nosema disease in honey bees are already chaotic, publishing another un-reliable study does not help find a field applicable treatment that can replace fumagillin.

The discussion is not well developed. I felt that I am grading an assignment from an undergraduate when I review this section. In a brief search, I found several facts about fenbendazole that have not been mentioned in this manuscript. For example, it has been used as an Encephalitozoon cuniculi (a well-studied microsporidium) treatment in rabbits (Suter et al., 2001), which is not included in this manuscript; but they did find treatment cases in fishes and silkworms of the other testing drugs. Fenbendazole possibility interferes with beta-tubulin polymerization as the drug function according to a review (Han and Weiss 2018), but the discussion of this manuscript stated there is no clue for the drug function.

I have not detailly searched every statement in the discussion. I thought the statements of fenbendazole should be fully researched since it was the best drug candidate found in this manuscript, but it was under-researched in this manuscript and quite disappointing. Fenbendazole is a broadband drug that might have toxicity effects on bees similar to that of fumagillin. This possibility should be well addressed in this manuscript, and I would want to find the discussion on the possibility of developing fenbendazole into a field treatment.

Several flaws were found in the M&Ms. The most obvious one is the solubilities of the testing drugs, which has greatly affected the reliabilities of the results. Most of the testing drugs are water-insoluble, noted as <100 mg/L in chemical properties, including curcumin and fenbendazole. In this manuscript, it stated all drugs were dissolved as 1000mg/L in water without any additional solvent, and then serial dilutions were made. Since the drug cannot be dissolved at the initial concentration, the dilutions are all problematic and possibly wrong in the trials. This may explain why there were no continuous effects in the trials comparing dilutions of drugs (Fig.6). Other flaws included the inoculation doses of nosema spores. They stated 2 million per bee in the results, but it was < 0.2 million in the M&Ms. The status of nosema spore storage was not clarified in the M&Ms, which may affect the spore viabilities. Other things that need to be clarified include the method of cleaning the reused cages, replacement of gravity feeders (the M&Ms said only added sugar water in the 20 days trial), and if any protein diets were given to the bees. Oddly, they stated that cage bees have many stressors, but they decided not to give any protein source in the diet, which adds nutrition stresses and development deficiencies.

There are many misc. things that I noted that are annoying and can affect the credibility of the manuscript, for example, some references have wrong information for the publishing journals. Please forgive me that I do not have time to list all the small things.

Han, Bing, and Louis M. Weiss. 2018. 'Therapeutic targets for the treatment of microsporidiosis in humans', Expert opinion on therapeutic targets, 22: 903-15.

Suter C, Müller-Doblies UU, Hatt JM, Deplazes P. Prevention and treatment of Encephalitozoon cuniculi infection in rabbits with fenbendazole. Vet Rec. 2001 Apr 14;148(15):478-80. doi: 10.1136/vr.148.15.478. PMID: 11334074.

Author Response

vivo Inhibitory Assessment of Potential Antifungal Agents on Nosema ceranae Proliferation

in Honey Bees 

Pathogens 1699168- Reviewer’s comments

Reviewer 1:

This manuscript is well-written and researched in the introduction and the abstract; however, things are getting worse in the following sections. The results are not impressive, the discussion is far worse than the introduction, and the M&Ms are problematic with some technique flaws. These findings made me wonder if the results reported in this manuscript are real facts. Flaws in M&Ms are not acceptable for papers that trying to identify drugs. Since the current understanding of research on alternative treatments for nosema disease in honey bees are already chaotic, publishing another un-reliable study does not help find a field applicable treatment that can replace fumagillin.

Many studies showed good efficacy of fumagillin for treating the Nosema-infected honey bees. However, reliance on treatment Nosema-infected colonies using fumagillin year after year may lead to developing resistance to this drug. In light of this issue, our goals in this project were to focus on screening alternative potential compounds already on the market, to provide alternative fungicides options to control of Nosema disease in honey bee colonies. Thus, by alternating use of the newly developed products with fumagillin, beekeepers will be able to mange resistance development to fumagillin and keep healthy bee in Canada. 

The discussion is not well developed. I felt that I am grading an assignment from an undergraduate when I review this section. In a brief search, I found several facts about fenbendazole that have not been mentioned in this manuscript. For example, it has been used as an Encephalitozoon cuniculi (a well-studied microsporidium) treatment in rabbits (Suter et al., 2001), which is not included in this manuscript; but they did find treatment cases in fishes and silkworms of the other testing drugs. Fenbendazole possibility interferes with beta-tubulin polymerization as the drug function according to a review (Han and Weiss 2018), but the discussion of this manuscript stated there is no clue for the drug function.

The effects of fenbendazole on the microtubule disorganization and its mode of action were already cited in the manuscript (for example Lines 361-363).   

More references associated with fenbendazole including Suter et al. (2001), and Han and Weiss (2018) are cited in the introduction (lines 110-116) and discussion (lines 360-362) sections. 

I have not detailly searched every statement in the discussion. I thought the statements of fenbendazole should be fully researched since it was the best drug candidate found in this manuscript, but it was under-researched in this manuscript and quite disappointing. Fenbendazole is a broadband drug that might have toxicity effects on bees similar to that of fumagillin. This possibility should be well addressed in this manuscript, and I would want to find the discussion on the possibility of developing fenbendazole into a field treatment.

We agree with the above point. Therefore, we suggested to expand this laboratory bioassay into the field scale to find the role of fenbendazole in Nosema control at colony level, and possibly risk of fenbendazole to the honey bee. This will also consider finding any residues in produced honey to determine the safety of honey for human consumption (lines 383-385). 

Several flaws were found in the M&Ms. The most obvious one is the solubilities of the testing drugs, which has greatly affected the reliabilities of the results. Most of the testing drugs are water-insoluble, noted as <100 mg/L in chemical properties, including curcumin and fenbendazole. In this manuscript, it stated all drugs were dissolved as 1000mg/L in water without any additional solvent, and then serial dilutions were made. Since the drug cannot be dissolved at the initial concentration, the dilutions are all problematic and possibly wrong in the trials. This may explain why there were no continuous effects in the trials comparing dilutions of drugs (Fig.6).

This a good point. To solve the active ingredients (AIs) and make the stock solutions, we followed our previous techniques (Bahreini et al 2020, 2021, 2022). To solve chemicals in 50% syrup (water-based), first we solved AIs in acetone to make stock solution, then the stock solution was added to a certain amount of sugar syrup to reach out the final dilution. To clarify this, we edited the M&M.

-Bahreini, R., M. Nasr, C. Docherty, O. de Herdt, S. Muirhead, and D. Feindel, 2020. Evaluation of potential miticide toxicity to Varroa destructor and honey bees, Apis mellifera, under laboratory conditions. Scientific Reports, 10: 21529.

-Bahreini R., M. Nasr C. Docherty, D. Feindel, S. Muirhead, O. de Herdt, 2021. New bioassay cage methodology for in vitro studies on Varroa destructor and Apis mellifera. PLoS ONE 16(4): e0250594.

-Bahreini R., M. Nasr C. Docherty, S. Muirhead, O. de Herdt, D. Feindel. 2022. Miticidal Activity of Fenazaquin and Fenpyroximate Against Varroa destructor, an Ectoparasite of Apis mellifera. Pest Management Science, 78: 1686-1697.

Other flaws included the inoculation doses of nosema spores. They stated 2 million per bee in the results, but it was < 0.2 million in the M&Ms. The status of nosema spore storage was not clarified in the M&Ms, which may affect the spore viabilities.

The number of Nosema spores per cage or per bee are corrected. The fresh Nosema spores were collected and used at the same day.    

Other things that need to be clarified include the method of cleaning the reused cages, replacement of gravity feeders (the M&Ms said only added sugar water in the 20 days trial), and if any protein diets were given to the bees. Oddly, they stated that cage bees have many stressors, but they decided not to give any protein source in the diet, which adds nutrition stresses and development deficiencies.

After the experiment, the used cages were double washed with hot water and soup using heavy duty dishwasher machine. The metal screens (under feeders) that were directly exposed to food, were replaced with new screen for future tests. For each trail, new tubes were used. The feeder tubes re-filled with fresh-make dilution of AIs, when cages needed more food (lines 528-533).

To avoid contamination with additional Nosema spores and residue of pesticides through commercial pollen patties, the bees were caged without pollen patties source. Therefore, the nutrition stress was equal for all treatments. M&M and discussion were edited (lines 527-528).            

 There are many misc. things that I noted that are annoying and can affect the credibility of the manuscript, for example, some references have wrong information for the publishing journals. Please forgive me that I do not have time to list all the small things.

 We revised the manuscript and added below references as well. Thank you.  

Han, Bing, and Louis M. Weiss. 2018. 'Therapeutic targets for the treatment of microsporidiosis in humans', Expert opinion on therapeutic targets, 22: 903-15.

Suter C, Müller-Doblies UU, Hatt JM, Deplazes P. Prevention and treatment of Encephalitozoon cuniculi infection in rabbits with fenbendazole. Vet Rec. 2001 Apr 14;148(15):478-80. doi: 10.1136/vr.148.15.478. PMID: 11334074.

Reviewer 2 Report

This study were to focus on in vivo screening of alternative compounds already on the market to control of Nosema disease in honey bees, and to determine the sublethal effects of candidate compounds on honey bee longevity. Thus, providing beekeepers with promising alternative anti-fungal compound(s) to integrate into their Nosema management plan while also reducing the risk of Nosema developing fumagillin-resistance. In this study, the biological activity of the active ingredient fenbendazole, on Nosema was investigated in honey bees. The results demonstrate that among tested compounds (curcumin, fenbendazole, nitrofurazone and ornidazole), fenbendazole showed the most promise as agent to control N. ceranae infections in honey bees under laboratory conditions. This research is promissory for the creation of new therapeutic agents to control Nosema infection in honey bees in the field.

I suggest minor revisions of the text.

  1. In the figure legends of each figure, the sentence “Means followed by same letter are not significantly different among treatments (Wilcoxon multiple comparison, p> 0.05).” it is better to say ““Means followed by different letter are significantly different among treatments (Wilcoxon multiple comparison, p< 0.05).”
  2. In section 2.2, it is better to explain the specific information of the positive control, negative control and reference control.
  3. In section 2.4, line188, what is F means?
  4. In section 4.2, line 419-420, I do not find the primers information. It is better to supply the specific information, so that others can reproduce the experiments.
  5. Since there are many technical terms in the manuscript, it is difficult to understand. It is better to try to make the paper more readable.

Author Response

In vivo Inhibitory Assessment of Potential Antifungal Agents on Nosema ceranae Proliferation

in Honey Bees 

Pathogens 1699168- Reviewer’s comments

Reviewer 2:

This study were to focus on in vivo screening of alternative compounds already on the market to control of Nosema disease in honey bees, and to determine the sublethal effects of candidate compounds on honey bee longevity. Thus, providing beekeepers with promising alternative anti-fungal compound(s) to integrate into their Nosema management plan while also reducing the risk of Nosema developing fumagillin-resistance. In this study, the biological activity of the active ingredient fenbendazole, on Nosema was investigated in honey bees. The results demonstrate that among tested compounds (curcumin, fenbendazole, nitrofurazone and ornidazole), fenbendazole showed the most promise as agent to control N. ceranae infections in honey bees under laboratory conditions. This research is promissory for the creation of new therapeutic agents to control Nosema infection in honey bees in the field.

I suggest minor revisions of the text.

Dear reviewer, some line numbers that you cited here are not match with the manuscript. We tried the best to apply your comments. 

-In the figure legends of each figure, the sentence “Means followed by same letter are not significantly different among treatments (Wilcoxon multiple comparison, p> 0.05).” it is better to say ““Means followed by different letter are significantly different among treatments (Wilcoxon multiple comparison, p< 0.05).” The captions of all figures and tables were edited.

-In section 2.2, it is better to explain the specific information of the positive control, negative control and reference control. More details were added to the section 2.2 for control treatments.

-In section 2.4, line188, what is F means? We couldn’t find “F” in the line 188 on the first submitted version.

-In section 4.2, line 419-420, I do not find the primers information. It is better to supply the specific information, so that others can reproduce the experiments. Sequences of the primers are added (lines 549-552). 

-Since there are many technical terms in the manuscript, it is difficult to understand. It is better to try to make the paper more readable. We tried to revise the introduction, results, discussion and materials and methods.

Reviewer 3 Report

Dear Editor,

The authors of this manuscript evaluate in vivo effects of various organic compounds on Nosema ceranae proliferation in honey bees.

I have some suggestions to improve quality of the manuscript.

Firsts of all I would like to advise authors to replace Nosema ceranae with Varimorpha (Nosema) ceranae.

Line 8. Nosema ceranae Fries – replace with Fries, 1996. Also in Lines 29-30.

Line 20. than non-treated Nosema-infested bees (20%) – The authors probably meant the treated with Fumagilin-B® bees! Please, explain?

Line 50-52. Please, remove this sentence. It is not associated with the content of the paper.

Line 61. Please, cite more resent papers:

Shumkova R, Balkanska R, Hristov P. The Herbal Supplements NOZEMAT HERB® and NOZEMAT HERB PLUS®: An Alternative Therapy for N. ceranae Infection and Its Effects on Honey Bee Strength and Production Traits. Pathogens. 2021; 10(2):234. https://doi.org/10.3390/pathogens10020234

Chen, X., Wang, S., Xu, Y., Gong, H., Wu, Y., Chen, Y., ... & Zheng, H. (2021). Protective potential of Chinese herbal extracts against microsporidian Nosema ceranae, an emergent pathogen of western honey bees, Apis mellifera L. Journal of Asia-Pacific Entomology, 24(1), 502-512.

Ptaszyńska, A.A.; Załuski, D. Extracts from Eleutherococcus senticosus (Rupr. et Maxim.) Maxim. roots: A new hope against honeybee death caused by nosemosis. Molecules 2020, 25, 4452.

Line 73-82. Please, give a short description of mentioned products – nitroimidazole, ornidazole etc.

Do the authors have information on whether resistance will be created against the drugs they use in the combat against nosematosis? The question of whether there will be an accumulation in bee products and what danger they would pose to human health also remains important?

Line 389. 53.54 °N, 113.49 °W – Please see the following format (41°35′7.01″ N, 24°41′30.98″ E).

Line 405. It is not clear how many colonies were used, how they were selected, what their healthy status was. Please, add an information.

Line 407. What RH does mean?

Line 411. and homogenized with a mortar and pestle in water (1 mL of water per ventriculus) – this approach is not suitable due to the fact that many DNase may digest DNA. Instead, TAE buffer is more appropriate.

Line 418. The products were quantified – replace with the purificated/extracted DNA.

Line 419-420. Please, give the sequences of the primers.

In fact the authors use multiplex PCR reactions with 4 primers. Please, add this information.

Line 429. newly-emerged worker bees – indeed these bees are house bees, before to become worker bees.

Line 507-509. For multiply measurement a Tukey's Test for Post-Hoc Analysis following Bonferroni correction must be apply.

In the Result section there is no information what Nosema spp. was detected???

The caption in Figure 1 not visualized different dilutions of tested compounds. Please, reformulate.

Also, it is more suitable for y axis in all figures to add Nosema spore loads.

Line 230-231. This is a wrong statement.

Line 241. newly-emerged worker bees – Please, see above explanation.

Author Response

In vivo Inhibitory Assessment of Potential Antifungal Agents on Nosema ceranae Proliferation

in Honey Bees 

Pathogens 1699168- Reviewer’s comments

Reviewer 3:

The authors of this manuscript evaluate in vivo effects of various organic compounds on Nosema ceranae proliferation in honey bees.

Dear reviewer, some line numbers that you cited here are not match with the manuscript. We tried the best to apply your comments. 

I have some suggestions to improve quality of the manuscript.

Firsts of all I would like to advise authors to replace Nosema ceranae with Varimorpha (Nosema)  ceranae. We decided still to use “Nosema” as genus name for Nosema pathogen instead of Varimorpha. However, this is cited in the introduction (lines 48-50).   

Line 8. Nosema ceranae Fries – replace with Fries, 1996. Also in Lines 29-30. Done (line 23).

Line 20. than non-treated Nosema-infested bees (20%) – The authors probably meant the treated with Fumagilin-B® bees! Please, explain? Done. “non-treated Nosema-infested bees” is the positive control (lines 36).

Line 50-52. Please, remove this sentence. It is not associated with the content of the paper. Done.

Line 61. Please, cite more resent papers: Below references are cited in the introduction.

Shumkova R, Balkanska R, Hristov P. The Herbal Supplements NOZEMAT HERB® and NOZEMAT HERB PLUS®: An Alternative Therapy for N. ceranae Infection and Its Effects on Honey Bee Strength and Production Traits. Pathogens. 2021; 10(2):234. https://doi.org/10.3390/pathogens10020234

Chen, X., Wang, S., Xu, Y., Gong, H., Wu, Y., Chen, Y., ... & Zheng, H. (2021). Protective potential of Chinese herbal extracts against microsporidian Nosema ceranae, an emergent pathogen of western honey bees, Apis mellifera L. Journal of Asia-Pacific Entomology, 24(1), 502-512.

Ptaszyńska, A.A.; Załuski, D. Extracts from Eleutherococcus senticosus (Rupr. et Maxim.) Maxim. roots: A new hope against honeybee death caused by nosemosis. Molecules 2020, 25, 4452.

Line 73-82. Please, give a short description of mentioned products – nitroimidazole, ornidazole etc. More statements are added to the introduction about candidate products (lines 94-123).

Do the authors have information on whether resistance will be created against the drugs they use in the combat against nosematosis? The question of whether there will be an accumulation in bee products and what danger they would pose to human health also remains important?

There is no enough information on the N. ceranae or N. apis resistance development to medications. Nevertheless, the periodic fumagillin treatment results in exposure of multiple generations of bees and pathogens to the drug. The use of this practice since mid 1950’s appears to provide an environment to develop fumagillin-resistant Nosema strains (Huang et al. 2013). So far, despite N. apis has been exposed to fumagillin for more than 70 years in North America, no reports or studies showed evidence of resistance to fumagillin. This issue is considered in the discussion section.     

Line 389. 53.54 °N, 113.49 °W – Please see the following format (41°35′7.01″ N, 24°41′30.98″ E). Done (line 438).

Line 405. It is not clear how many colonies were used, how they were selected, what their healthy status was. Please, add an information. Information about the experimental source colonies are added in the Materials and Methods section (lines 429-436).

Line 407. What RH does mean? Relative Humidity is cited (line 460).

Line 411. and homogenized with a mortar and pestle in water (1 mL of water per ventriculus) – this approach is not suitable due to the fact that many DNase may digest DNA. Instead, TAE buffer is more appropriate. Bees were homogenized in water to count Nosema spores for microscopic method. We agree to use TAE buffer in PCR technique and it will be considered in future research. Thank you.  

Line 418. The products were quantified – replace with the purificated/extracted DNA. The words are edited (lines 547-548).

Line 419-420. Please, give the sequences of the primers. Done (lines 548-552).

In fact the authors use multiplex PCR reactions with 4 primers. Please, add this information. Done (line 545).

Line 429. newly-emerged worker bees – indeed these bees are house bees, before to become worker bees. This word is edited through the manuscript.

Line 507-509. For multiply measurement a Tukey's Test for Post-Hoc Analysis following Bonferroni correction must be apply. In this study, we used Bonferroni for Post Hoc analyses instead of Tukey’s, due to the Bonferroni method is more rigorous than the Tukey test, which produces the narrowest confidence intervals than other method.

In the Result section there is no information what Nosema spp. was detected???

It was mentioned in the first paragraph of Result section (lines 137-138).

The caption in Figure 1 not visualized different dilutions of tested compounds. Please, reformulate. Done.

Also, it is more suitable for y axis in all figures to add Nosema spore loads. Done.

Line 230-231. This is a wrong statement. We do not know what exactly lines 230-231 are. In lines 230-231, the Nosema distribution in the colony via trophallaxis behavior is discussed. Do you mean this statement is wrong?

Line 241. newly-emerged worker bees – Please, see above explanation. Done.

Reviewer 4 Report

The manuscript submitted by Bahreini and colleagues aimed to investigate antifungal substances against N. ceranae in experimentally infected honey bees.

The research is very interesting and, at this moment, it was very helpful to have information about natural substances against nosemosis. 

In my opinion, the research is well conducted and the results seem promising.

However, in the text are present a lot of typos and a few grammatical mistakes. I encourage a careful reading of the manuscript.

Finally, considering the point of the paper aimed at the use of natural substances against N. ceranae infection, I encourage you to report these references, including them in the view of natural strategy to control nosemosis. They are only examples, but there are some other references unreported.
-Borges  et al. (2021, Microorganisms 9, 481)
-El Khoury et al. (2018, Front. Ecol. Evol. 6, 58)
-Nanetti et al. (2021, Microorganism 9, 949)
- Cilia et al. (2020, Vet. Sci 7, 125)
- Ugolini et al. (2021, Biomolecules 11, 1657)

Author Response

In vivo Inhibitory Assessment of Potential Antifungal Agents on Nosema ceranae Proliferation

in Honey Bees 

Pathogens 1699168- Reviewer’s comments

Reviewer 4:

The manuscript submitted by Bahreini and colleagues aimed to investigate antifungal substances against N. ceranae in experimentally infected honey bees.

The research is very interesting and, at this moment, it was very helpful to have information about natural substances against nosemosis. 

In my opinion, the research is well conducted and the results seem promising.

However, in the text are present a lot of typos and a few grammatical mistakes. I encourage a careful reading of the manuscript. The manuscript is revised and edited.

Finally, considering the point of the paper aimed at the use of natural substances against N. ceranae infection, I encourage you to report these references, including them in the view of natural strategy to control nosemosis. They are only examples, but there are some other references unreported. Relevant articles including below references are cited in the introduction and discussion sections.

-Borges  et al. (2021, Microorganisms 9, 481)

-El Khoury et al. (2018, Front. Ecol. Evol. 6, 58)

-Nanetti et al. (2021, Microorganism 9, 949)

-Cilia et al. (2020, Vet. Sci 7, 125)

-Ugolini et al. (2021, Biomolecules 11, 1657)

Round 2

Reviewer 3 Report

The authors significantly improved their paper and I suggest to accept in present form.

Author Response

Dear Reviewer 3,

Thank you for spending your time to make a review on our manuscript.

Regards

Authors  

hank you again for spending your time to make a review